



# A modelling study of an extreme rainfall event along the northern coast of Taiwan on 2 June 2017

Chung-Chieh Wang[1], Ting-Yu Yeh[1], Ming-Siang Li[1], Kazuhisa Tsuboki[2], Ching-Hwang Liu[3]

[1]Department of Earth Sciences, National Taiwan Normal University, Taipei, 11677, Taiwan
[2]Institute for Space-Earth Environmental Research, Nagoya University, Nagoya, 464-8601, Japan
[3]Department of Atmospheric Sciences, Chinese Culture University, Taipei, 11114, Taiwan

*Correspondence to*: Chung-Chieh Wang (cwang@ntnu.edu.tw)

**Abstract.** In this study, the extreme rainfall event on 2 June 2017 along the northern coast of Taiwan is studied from a modeling perspective. While a peak amount of 645 mm was observed, two 1-km experiments produced about 400 and 541
mm, respectively, using different initial and boundary conditions, and thus are compared to isolate the key reasons for a higher total amount in the second run. While the conditions in frontal intensity and its slow movement are similar in both runs, the frontal rainband remains stationary for a long period in this second run due to a frontal disturbance that acts to enhance the pre-frontal southwesterly flow and focus its convergence with the post-frontal flow right across the coastline. Identified as the key difference, this low-pressure disturbance is supported by the observation, and without it in the first run,
multiple slow-moving rainbands pass through the coastal region and produce more widely spread but less concentrated rainfall, resulting in the lower peak amount by comparison.

To explore and test the effects of Taiwan's topography in this event, three 3-km runs are also used. It is found that the removal of the terrain in northern Taiwan makes only minor differences, in contrast to the result of a recent study. Only when the entire island topography of Taiwan is removed, does the result show significant differences. In this case, the
blocking and deflecting effects on the pre-frontal flow are absent, and the heavy rainfall in northern Taiwan does not occur.

## 1 Introduction

During the transition period from the northeastern to southwestern monsoon, there exists an early-summer rainy season in many regions in East Asia, including China, Taiwan, Japan, and Korea (e.g., Lau et al., 1988; Ding, 1992; Chen, 2004; Ding and Chan, 2005). Known as the Mei-yu (plum rain) season in Taiwan, it is defined as May and June by the Central Weather
Bureau (CWB). During this season, the Mei-yu front often forms repeatedly between the cold continental and warm maritime air masses (located in China and the subtropical western North Pacific, respectively) and moves in to affect the region, and subsequently becomes stationary to bring about the continuous rainy conditions (Chen and Chi, 1980; Chen 1992; Ding and Chan, 2005). When a Mei-yu front develops or approaches Taiwan, the horizontal pressure gradient can strengthen at times and the wind speed south of the front increases to form a low-level jet (LLJ; e.g., Chen et al., 1994; Chen and Chen,
1995). Coming from the southwest, the LLJ can often transport the moist and unstable air toward Taiwan and the Mei-yu



front from the upstream ocean, and thus has long been recognized as an important feature to cause heavy rainfall in Taiwan in the literature (e.g., Chen and Yu, 1988; Kuo and Chen, 1990; Chen et al., 2005, 2008). Under such conditions, not only organized mesoscale convective systems (MCSs) such as squall lines (Rotunno et al., 1988; Houze et al., 1989; Lin et al., 1990; Jou and Deng, 1992; Chen and Chou, 1993; Wang et al., 2014a) can develop near the front and make landfall in

Taiwan (e.g., Kuo and Chen, 1990; Chen, 1992, 2004; Chen et al., 1998; Wang et al., 2011; Xu et al., 2012;), but the steep topography of Taiwan, with the highest peak reaching almost 4 km (Fig. 1a), also act to enhance the convection or trigger new convection by forced uplift (e.g., Kuo and Chen, 1990; Nagata and Ogura, 1991; Lin, 1993; Lin et al., 2001; Wang et al., 2022b). Therefore, the long-term Mei-yu rainfall climatology shows two prominent centers over the mountain interiors in central and southern Taiwan, respectively (Chi, 2006; Wang et al., 2022b), and they even also appear in total rainfall in most

of individual seasons (e.g., Yeh and Chen, 1998; Chien and Jou, 2004; Wang et al., 2017, 2022a). A third but less pronounced center (Chi, 2006) appears in northern Taiwan (and in some seasons) and is typically associated with the Mei-yu front.

In addition to forced uplift, the topography of Taiwan also has thermodynamic effects and is an important contributor to the island circulation and diurnal cycle of rainfall during the Mei-yu season (e.g., Akaeda et al., 1995; Chen et al., 1999; Kerns

et al., 2010; Johnson, 2011; Ruppert et al., 2013; Wang et al., 2014b, 2022b). It also exerts a significant blocking effect on the oncoming environmental prevailing flow (e.g., Pierrehumbert and Wyman, 1985; Banta, 1990; Yeh and Chen, 2002; Wang et al., 2005). In the latter situation, Yeh and Chen (2003) suggested that the deflection of southwesterly flow by the Central Mountain Range (CMR) of Taiwan can often produce a local barrier jet (BJ) off the northwestern coast of the island (also Li and Chen, 1998; Yeh and Chen, 2002). The low-level convergence induced by this BJ can lead to heavy rainfall in

the area (Yeh and Chen, 2003) when a frontal rainband also arrives, thus contributing to the third rainfall center in northern Taiwan.

During the past two decades or so, only two events reached 500 mm in 24 h (defined as "extremely heavy rainfall" by the CWB, the criterion to have one day off work and school) in northern Taiwan in the Mei-yu season, on 11-12 June 2012 and on 2 June 2017, respectively. With serious flooding in or close to the populous Taipei metropolitan area (Fig. 1b), each event

caused severe property damage and economic loss, and thus demand particular attention from the research community. In the 11-12 June event in 2012, Taipei received a peak rainfall of 510 mm in 24 h, caused by two successive rainbands overnight: a prefrontal squall line and a stationary rainband that formed in northern Taiwan Strait and extended into northern Taiwan (Wang et al., 2016). While each rainband lasted for about 6 h, the second one was studied in detail and found to form ahead (south) of the surface front over the northern Taiwan Strait, along the convergence zone between the southwesterly flow

deflected by the topography and the unblocked west-southwesterly flow farther offshore in the environment (Wang et al., 2016). Contributing toward the vigor of the convection and thus total rainfall, the back-building process occurring inside the rainband without the presence of the cold pool was also studied. On the other hand, Chen et al. (2018) emphasized on the high moisture content and moisture flux inside the marine boundary layer in this event. In their sensitivity test with the topography of Taiwan removed, the BJ did not form offshore of northwestern Taiwan (also Ke et al., 2019). Consequently,



without the rainband between the BJ and the environmental flow, only a fraction of the observed rainfall was produced in
northern Taiwan (Chen et al., 2018).

In the second event on 2 June 2017, the rainfall amount was even higher (645 mm in 24 h) and was maximized along the
coast at the northern tip of Taiwan, caused by a single intense, quasi-stationary, and long-lasting rainband along the Mei-yu
front (more details in Section 3). For this event, Wang et al. (2021, hereafter referred to as WLC21) performed an ensemble-

based sensitivity analysis (ESA, Ancell and Hakim, 2007; Torn and Hakim, 2008; Bednarczyk and Ancell, 2015) using 45
forecast members at grid sizes ($\Delta x$) of 2.5-5 km. In a quantitative way, the study identified several factors influencing the
areal-mean rainfall amount over a 6-h period inside an area (roughly $80 \times 55$ km$^2$ in size) centered at the northern tip of
Taiwan. Expressed as their impact on rainfall every increase in one standard deviation (SD), these factors are: (1) surface
frontal position and moving speed ($-16.00$ mm per 5 km h$^{-1}$), (2) position of 700-hPa wind-shift line ($+12.59$ mm per 0.4°

latitude), (3) environmental moisture amount (mixing ratio) near the surface front ($+11.73$ mm per 0.92 g kg$^{-1}$), (4) timing
and location of frontal low-pressure disturbance ($+11.03$ mm per 1.38° longitude), and (5) frontal intensity ($+9.58$ mm per 3
K in equivalent potential temperature difference across 0.5°). While many of these factors are interconnected, the local
rainfall tended to be higher in ensemble members if the duration of heavier rainfall is longer and the near-surface
convergence across the front and over the upstream area (along the northwestern coast of Taiwan) is stronger, as expected

(WLC21). Among the 45 members, the best one (named M18) produced 360 mm in 12 h at the northern coast. More recently,
Tu et al. (2022) also studied this 2017 event, and attributed the coastal heavy rainfall to the postfrontal cold air being too
shallow to climb over the mountain in northern Taiwan (~1.1 km in peak elevation, cf. Fig. 1b) for the first 8 h. While the
maximum value is not explicitly noted, their 3-km control simulation produced likely around 450 mm in 24 h (for 2 June in
LST) at the northern coast (their Fig. 18b), compared to 645 mm in the observation. When the terrain in northern Taiwan

was removed (lowered to $\leq 150$ m) in their sensitivity test, the local rainfall center shifted to northwestern coast and the peak
amount reduced to below 400 mm, likely around 350 mm (their Fig. 18c), thus perhaps by around 100 mm or 25%.

A few questions remain regarding this event on 2 June 2017. First, it appears quite challenging to reproduce a peak amount
close to the observation (645 mm) at the correct location at $\Delta x$ of 2.5-3 km, including Lupo et al. (2020) and Chung et al.
(2020), especially in forecasts. In WLC21, all ensemble members had 12-h peak rainfall of 360 mm or less, and a relatively

low ensemble mean value (~130 mm) and a large spread suggest a low predictability for the extreme rainfall in the northern
coast of Taiwan in this event. So, can a rainfall distribution closer to the observation be obtained, say, reaching "extremely
heavy rainfall" ($\geq 500$ mm), perhaps using a higher resolution? Second, the five factors identified by WLC21 using ESA are
important to differentiate the more-rainy members (with peak 12-h amounts of about 150-350 mm) from less-rainy ones
(below 150 mm) in northern Taiwan. Obviously, such peak amounts still differ quite a lot from the observation, and thus

lead to our second question: Do certain factors among the five become dominant over the others, if the model is to produce a
peak amount of over 500 mm at the northern coast? In this study, we seek answers to the above questions, and both will be
shown to be affirmative. In one of our 1-km experiment, a peak 24-h rainfall of 541 mm is captured along the northern coast,





while another experiment has only close to 400 mm. Thus, these two model runs are compared in detail to isolate the reasons

for the considerably higher peak amount in the former experiment. In contrast to Tu et al. (2022), the key difference found

here is not the existence of the topography in northern Taiwan, and the related aspects are discussed and elaborated later. For

this purpose, results from a few 3-km experiments are used.

The remaining part of this paper is arranged as follows. The data, numerical model, and experiments are described in Section

2. In Section 3, an overview of the case on 2 June 2017 is given. The results of our 3-km experiments are discussed in

Section 4, including the tests on topographic effects. In Section 5, the 1-km experiments are presented and contrasted to

isolate the key differences in the model for a peak rainfall amount approaching the observed value along the northern coast.

Further discussion is given in Section 6, followed by the conclusion and summary in Section 7.

## 2 Data and methodology

### 2.1 Data

The observational data used in this study include weather maps, sounding data, rain-gauge data, merged rainfall estimates

from radar and gauge observations from the Central Weather Bureau (CWB) of Taiwan, and also the gridded analyses from

the National Centers for Environmental Prediction (NCEP) and the Navy Global Environmental Model (NAVGEM), Naval

Research Laboratory of the USA. The weather maps and sounding data at Panchiao (near Taipei) are used for the discussion

in synoptic environment and thermodynamic conditions, and the hourly rainfall data (Hsu, 1998) and the Quantitative

Precipitation Estimation and Segregation using Multiple Sensors (QPESUMS), a radar-derived estimates calibrated by rain

gauges over land (Gourley et al., 2001), at 10-min intervals are used for the overview of the stationary rainband and extreme

rainfall in the present event.

The analysis and discussion in this study is also aided by the use of gridded global analyses during our case period. These

datasets include the final (FNL) analyses from the NCEP Global Forecast System (GFS) every 6 h, at 0.25° × 0.25° and 26

levels (at surface and from 1000 to 10 hPa, Kalnay et al., 1990; Moorthi et al., 2001; Kleist et al., 2009), as well as the

NAVGEM (version 1.4, T425L50; Metzger et al., 2013). Both analyses include all the important variables, such as pressure

($p$), temperature ($T$), geopotential height ($z_\Phi$), horizontal wind components ($u$ and $v$), and moisture content, and the former

are also used to drive our 3-km experiments (described later in Section 2.3). Where needed, both the observational data and

gridded analyses are used to compare with and validate the model results.

### 2.2 The CReSS model

The numerical model used in this study is the Cloud-Resolving Storm Simulator (CReSS, version 3.4.2) developed at the

Nagoya University, Japan (Tsuboki and Sakakibara, 2002, 2007). This is a single-domain cloud model that employs a

nonhydrostatic and compressible equation set, and a terrain-following vertical coordinate system. As shown in Table 1, all

clouds are explicitly simulated in CReSS using a double-moment bulk cold-rain microphysical scheme with six species of





vapor, cloud water, cloud ice, rain, snow, and graupel. Parameterized sub-grid scale processes include turbulent mixing in
the planetary boundary layer, radiation, and surface momentum and energy fluxes with a substrate model (Table 1). Further
details regarding the CReSS model can be found in some earlier studies (e.g., Wang et al., 2012; 2014a,b, 2016, WLC21), or
online (http://www.rain.hyarc.nagoya-u.ac.jp/~tsuboki/cress_html/index_cress_eng.html).

### 2.3 Numerical experiments

A total of five experiments were performed and used in this study. Three of them are at a horizontal grid size of 3 km, and
employed the NCEP GFS FNL as their initial and boundary conditions (IC/BCs) for the 5-day period starting from 0000
UTC 30 May 2017 (Table 2). Using real topography data on a (1/120)° grid (about 900 m), the simulation S3 serves as the
control experiment and produced a peak 24-h rainfall of 437 mm in northern Taiwan (details in Section 4). Identical to S3 in
all other aspects, two experiments were designed to test the impact of topography in the present event: The S3-NT in which
the topography of the entire Taiwan is removed, and S3-NNT where only the terrain in northern Taiwan is removed (Table
3). In Figs. 1a and 1b, the respective regions of terrain removal are shown, and any topography exceeding 1 m is set to 1 m
inside them in each test.

As the maximum rainfall at the northern coast of Taiwan in S3 was still about 200 mm below the observation, a 1-km
simulation named S1 was carried out to examine if more rainfall can be yielded using a finer horizontal grid (Table 3).
However, this run only had 393 mm along the coast, and thus did not produce any more rainfall over land. Interestingly,
another experiment F1 that used the forecast outputs of member M18 of WLC21 as the IC/BCs, with otherwise identical
setting (Table 3), was able to produce a peak amount of 541 mm along the northern coast, closer to the observation.
Therefore, these two experiments are examined and compared in more detail in Section 5 to isolate the key differences
between them that lead to the considerably different amounts of peak rainfall accumulation along the northern coast.

### 3 Case overview

#### 3.1 Synoptic and pre-frontal environment

The extreme-rainfall event in northern Taiwan during 1-2 June in the Mei-yu season of 2017 is briefly reviewed in this
section. First, the CWB surface weather maps overlaid with the NAVGEM 925-hPa flow fields surrounding Taiwan every 6
h from 1200 UTC 1 to 0600 UTC 2 June 2017 are shown in Fig. 2. A stationary surface Mei-yu front, with roughly an east-
west alignment, was located about 150 km north of Taiwan at 1200 UTC (or 2000 LST, where LST = UTC + 8 h) 1 June
(Fig. 2a). Near the commencement of heavy rainfall, it approached the northern coast of Taiwan around 1800 UTC (0200
LST 2 June, Fig. 2b). Afterward, the front slowly moved through the northern part of the island during 0000-0600 UTC
(0800-1400 LST) 2 June (Figs. 2c,d). Thus, the surface front remained in the vicinity of northern Taiwan for about 12 h. To
the south of the front, persistent southwesterly flow appeared at 925 hPa throughout this period (Fig. 2).



Figure 3 shows the synoptic conditions aloft in the troposphere at 1200 UTC 1 June. Extending from the low pressure over the Sea of Japan, the front (or trough) over the East China Sea and South China was at about 27.5°N at 850 hPa (Fig. 3a) and further north near 28.5°N at 700 hPa (Fig. 3b), with clearly easterly flow to its north. Further up at 500 hPa, the wind shift line became less apparent (Fig. 3c). Nonetheless, a baroclinic structure of the front was evident in Fig. 3, with a northwestward tilt with height. Associated with this, a veering of the strong flow (roughly 30 kts) near Taiwan existed, from west-southwesterly flow at 850 and 700 hPa to westerlies at 500 hPa, and further to west-northwesterlies at 200 hPa, indicating warm air advection as well.

The sounding observation made at Panchiao (near Taipei) in northern Taiwan at 1200 UTC 1 June (Fig. 4) showed a prefrontal environment that was well mixed in the PBL below 900 hPa in the early evening (2000 LST), consistent with the strong vertical wind shear near the surface (and Figs. 2a and 3). Also with gradual veering, the flow south of the surface front increased to 40 kts in speed at 900-850 hPa and further to 50 kts at 500 hPa, clearly reaching the criteria of the LLJ (Jou and Deng, 1992; Wang et al., 2014a). Above the PBL, the temperature lapse rate suggested conditional instability up to about 540 hPa (Fig. 4). The convective available potential energy (CAPE) of a surface air parcel was 576 J kg$^{-1}$, the convective inhibition (CIN) was about 64 J kg$^{-1}$, while the level of free convection (LFC) was relatively high, near 3.2 km at 692 hPa (Fig. 4). These parameters and the overall thermodynamic structure were very similar to those prior to the event on 11-12 June 2012 (583 and 78 J kg$^{-1}$, and 780 hPa) reviewed in Section 1 (Wang et al., 2016), and sufficient to support deep convection with enough forcing to trigger free ascent. Thus, with instability and forcing provided by the approaching Mei-yu front, deep convection developed and organized into a severe rainband, as described below in the next subsection.

**3.2 Stationary rainband and extreme rainfall along the northern coast of Taiwan**

In Fig. 5, hourly QPESUMS data (Gourley et al., 2001) near Taiwan, overlaid with the NCEP FNL surface horizontal winds, are shown from 1800 UTC 1 to 0500 UTC 2 June 2017 to depict the evolution of the intense rainband associated with the Mei-yu front. The frontal rainband first reached the northern tip of Taiwan around 1800 UTC 1 June (Fig. 5a), but remained stationary until at least 0200 UTC 2 June 2017 (Fig. 5i). Only afterward, it started to move inland slowly toward the south (Figs. 5j-l). Therefore, the rainband stayed along the northern coast of Taiwan for some 9-10 h overall, with roughly an east-west orientation throughout this period. The rainrate estimates along the northern coast were often 50-90 mm h$^{-1}$ and intense, thus leading to the extreme rainfall. During the heavy-rainfall period, the QPESUMS also indicated that there were few other rainbands near northern Taiwan, while the mountain regions in central and southern Taiwan also received persistent rainfall (Fig. 5). The ERA-5 analyses at 1000 hPa revealed a steady southwesterly flow of around 10 kts in the upstream region over the period, and the flow was deflected by the steep terrain of Taiwan at the windward side. The front (or wind shift line) in the FNL analyses mostly lagged the rainband in the QPESUMS by about 50-100 km, and thus was likely too far north.

Produced by the rainband seen in Fig. 5, the observed 24-h accumulated rainfall over Taiwan from 1600 UTC 1 to 1600 UTC 2 June (0000-2400 LST) reached 645 mm right along the northern coast (Fig. 6a). Two other rainfall centers also appeared along the CMR in central and southern Taiwan, each exceeding 300 mm. In fact, out of the 645 mm in northern



Taiwan, 635 mm of rainfall occurred within 12 h between 1600 UTC 1 and 0400 UTC 2 June (cf. Fig. 5, also WLC21), causing serious inundation and economic loss along the northern coast.

## 4. Results of 3-km experiments

### 4.1 Control simulation of S3

As described in Section 2.3, three simulation experiments were performed at $\Delta x = 3$ km, all driven by the NCEP GFS FNL analyses as IC/BCs with a fairly large domain and an initial time ($t_0$) at 0000 UTC 30 May 2017 (Table 2). In the control simulation S3, where the real topography of Taiwan was used, maximum 24-h rainfall of 438 mm was produced just offshore of northern Taiwan, with a peak amount of 437 mm over land (Fig. 6b), comparable to the simulation of Tu et al. (2022). However, the surface front in S3 arrived at northern Taiwan too early, by about 9 h (Fig. 7a), and thus the accumulation period in Fig. 6b (also Figs. 6c,d) was from 0700 UTC 1 June. The simulated front nevertheless remained stationary for roughly 10 h as in the observation (cf. Fig. 5), and only started to move south prior to 1900 UTC (Figs. 7b,c). While the timing error is not ideal but acceptable during the third day into the simulation, the S3 experiment is considered successful since the behavior of the front and rainband and the rainfall area were all in good agreement with the observations.

### 4.2 Sensitivity tests of S3-NT and S3-NNT

To test and clarify the role played by the topography, two sensitivity experiments were carried out to compare with S3, with identical setting except for the terrain height as described (Table 3). When the topography in northern Taiwan was removed in S3-NNT (cf. Fig. 1b), a maximum 24-h rainfall of 427 mm occurred right at the northern tip of Taiwan (Fig. 6c). Thus, a reduction of merely 10 mm with a slight shift in location was resulted. Likewise, the evolution of the front and rainband in S3-NNT (Figs. 7d-f) only exhibited small differences compared to those in S3 (cf. Figs. 7a-c). Therefore, this sensitivity test suggests that the terrain in northern Taiwan had only minor impact to rainfall accumulation, and is in contrast to the conclusion of Tu et al. (2022). At the very least, our results from S3 and S3-NNT indicates a greater variability and uncertainty in model simulations than assumed by Tu et al. (2022), and the relative height of the cold air behind the front to the topography was not the major factor determining the location and amount of peak rainfall over northern Taiwan in some experiments (like ours).

When the entire topography of Taiwan was removed, significant differences were obtained in S3-NT relative to S3 (Table 3). In this test, the peak 24-h rainfall offshore northern Taiwan is only 204 mm and that along the coast is not even 100 mm (Fig. 6d), similar to the result of Chen et al. (2018) for the event on 11-12 June 2012. Not only is the rainfall surrounding northern Taiwan greatly reduced, but the rainfall centers in the mountains also disappear altogether. This is because without the terrain, the near-surface (and low-level) southwesterly winds can blow across the flattened island without the blocking effect (Figs. 7g-i), Without deflection and convergence, the southwesterly flow over the Taiwan Strait during the event is weaker, thereby allowing the northerly flow to advance more rapidly. Also, the daming and southward intrusion of post-frontal cold





air along the eastern coast of Taiwan in S3-NT does not occur (Figs. 7g-i), in contrast to both S3 and S3-NNT. Interestingly, the surface front in S3-NT also stays near northern Taiwan for approximately the same length in time, but the rainfall is less

intense and less persistent. This result suggests that the local convergence at the front (and rainband) between the southwesterly and northeasterly flow, with the former being enhanced by the blocking effect of the island, was important to bring the rainfall along the northern coast up to a value over 400 mm in our 3-km experiments.

## 5. Results of 1-km experiments

### 5.1 Two contrasting experiments of S1 and F1

As mentioned, two experiments at a grid size of 1 km were performed and contrasted in this section. The first one is S1, driven by the outputs of S3 at 1-h intervals and starting from 2200 UTC 31 May 2017 for 30 h (Table 3), and it produced only 393 mm (to be discussed soon) in 24 h along the northern coast. On the other hand, a second experiment named F1 used the hourly outputs of M18 of WLC21 as IC/BCs and started from 1300 UTC 1 June (also for 30 h), but reached 541 mm in northern Taiwan. As mentioned, M18 ($\Delta x = 3$ km) yielded 360 mm of rainfall (in 12 h), the most among all 45 members in

WLC21. It was identical to S3 in model configuration and physical package (including cloud microphysics), and the two differed only in their IC/BCs: the NCEP GFS FNL analyses (0.25°, every 6 h) were used in S3 (cf. Table 2), while the GFS real-time gross analysis and forecasts (0.5°, every 3 h) were used in M18 (WLC21). Nevertheless, F1 produced almost 150 mm more rainfall than S1 at the northern coast of Taiwan, and to our knowledge, the value of 541 mm is also the closest to the observation from any model result for this event. Thus, to investigate and clarify the reasons between the considerably

different peak amounts of accumulated rainfall along the northern coast, the two experiments of S1 and F1 are examined and compared in detail in this section.

### 5.2 Frontal movement and difference in rainfall characteristics

The model-simulated surface frontal positions at 2-h intervals are first shown in Fig. 8 to examine whether there are significant differences in the frontal moving speed between the two 1-km experiments. In S1, the surface front reached the

northern tip of Taiwan at around 0200 UTC and remained at the northernmost part of the island until about 1500 UTC (Fig. 8a), and thus was stationary for around 13 h in total. On the other hand, the surface front in F1 moved through the same short distance near the northern coast in about 10 h, roughly from 1600 UTC 1 to 0200 UTC 2 June (Fig. 8b), in close agreement with Fig. 5. Even though somewhat shorter in duration of frontal stagnation, F1 produced more rainfall along the northern coast of Taiwan than S1. Thus, while WLC21 identified the timing and speed of frontal movement as an important factor to

an increased mean rainfall in northern Taiwan from their ESA, it does not appear to be as critical here when the peak rainfall reaches around 400 mm in both runs.

The modeled 24-h rainfall distributions in S1 and F1 are plotted and compared in Fig. 9, with the accumulation period starting from 0000 UTC 1 June for S1 and 1300 UTC 1 June for F1, respectively. Immediately apparent is that the rainfall in





S1 is more widespread, with much larger areas offshore and to the northwest of northern Taiwan receiving over 150-200 mm
(Fig. 9a), but a lower peak amount overland at 393 mm. On the contrary, the rainfall in F1 is much more concentrated right
around the northernmost part of the island (Fig. 9b), with a much smaller area receiving over 200 mm but higher peak
amounts, reaching 541 mm overland as mentioned and 618 mm over the ocean about 15 km offshore from the northern tip. If
12 h is used for accumulation, then the peak values are 576 (offshore) and 457 mm (on land) in F1 and 269 mm (on land) in
S1, respectively. As depicted in Fig. 9, three rectangular domains are chosen to compute the areal-mean rainfall, with a size
of $1.4° \times 0.8°$ ($x \times y$, domain L), $0.7° \times 0.4°$ (domain M), and $0.45° \times 0.2°$ (domain S), respectively, and the value computed
for the full domain and only the land portion (inside the domain) are given in Table 4. In domain L, overall the S1 run
produced slightly more total rainfall (219.86 mm) than F1 (213.42 mm), but the opposite is true as the averaging domain is
decreased in size and more focused on the northernmost part of Taiwan (Table 4). Inside the smallest domain (domain S), F1
had significantly more total rainfall (346.36 mm) than S1 (251.32 mm). If only the land portions are considered, then F1
consistently yielded more rainfall than S1 inside the three selected domains, but the difference becomes increasingly large as
the domain becomes smaller. Outside domain L, it is evident in Fig. 9 that S1 produced considerably more rainfall than F1
around the northern Taiwan Strait. Thus, Fig. 9 and Table 4 confirm that the rainfall in F1 is more concentrated over a
smaller region right at the northernmost part of Taiwan, but this is not the case in S1.

Next, the hourly rainfall at all model grid points in S1 and F1 inside each of the three domains are classified based on their
intensity, into seven groups of 0.01-1, 1-5, 5-10, 10-20, 20-50, 50-100, and ≥100 mm h$^{-1}$, respectively, and their fractions
against time are plotted in Fig. 10 to allow for an inspection on the temporal evolution of rainfall at different intensity ranges
between the two experiments. In domain L, except for a higher fraction (larger region) of little or no rainfall (< 0.01 mm h$^{-1}$)
in S1 during the first 12 h of the period shown (Fig. 10a), the intensity groups (higher ones toward the bottom) appear to be
comparable in terms of their fraction (percentage) and evolution in time between the two runs (Figs. 10a,b). As the domain
size decreased, it becomes increasingly apparent that a higher fraction of more intense rainfall (say, ≥20 mm h$^{-1}$) existed in
F1 compared to S1 during the heavy-rainfall period (Figs. 10c,d), especially inside the smallest domain that focuses on the
northernmost part of Taiwan (Figs. 10e,f). Also, the heavy rainfall in domain S is more persistent in F1, but tends to be
intermittent and only concentrate in a few periods of about 3-4 h in S1. In other words, the intense rainfall occurs in S1 in
pulsation but continuous in F1 over the northern coast, and therefore allows for a considerably higher local accumulation
amount in the latter model experiment (cf. Fig. 9). The underlying reasons for this difference is further explored and
discussed below.

**5.3 Location and evolution of rainbands**

In order to examine the location and evolution of rainbands associated with the front near northern Taiwan in the two 1-km
model experiments, hourly rainfall (ending at the indicated time) every 2 h in S1 and F1 are shown in Figs. 11 and 12,
respectively. In Fig. 11 for S1, the three more intense rainfall periods over the northern coast of Taiwan in Fig. 10e can be



identified: approximately during 0500-0900 UTC (Figs. 11a,b), 1100-1500 UTC (Figs. 11d,e), and around 1900-2000 UTC (Fig. 11h) on 1 June. While their moving speed may be slow, these rainbands indeed move continuously with time, across the northern coast of Taiwan in a successive manner (Fig. 11). At almost all the instances shown in Fig. 11, multiple rainbands near the front appear in S1 (cf. Fig. 8a), including the northern Taiwan Strait. In Fig. 12, on the other hand, a

different scenario is seen in the F1 experiment: the northern coast of Taiwan receives heavy rainfall more or less continuously, roughly from 1600 UTC 1 (Fig. 12b) to 0400 UTC 2 June (Fig. 12h) over a period of 12 h, consistent with Fig. 10f. This is because a local rainband in Fig. 12 forms between the prefrontal westerly or southwesterly winds (immediately offshore of northwestern Taiwan) and the cold northeasterly winds (north and northeast of Taiwan), right across the northern coast, and persists through much of this 12 h period in F1 (Figs. 12b-g).

Using plots like those in Figs. 11 and 12, hourly positions of rainbands around northern Taiwan in S1 and F1 were identified and contrasted in Fig. 13. Again, as old rainbands gradually move south after passing through the northern coast in S1, new bands form over the northern Taiwan Strait or north of Taiwan, and then approach and produce rainfall along the coastal region again (Figs. 13a-c). Between 0600 and 2200 UTC 1 June, at least three different rainbands affect the northern coast in Figs. 13a-c with gaps in between (cf. Fig. 10e), thus producing widespread rainfall but a lower peak amount in S1 (cf. Fig.

9a). On the other hand, a single stationary rainband persists for a long time (of over 10 h) right across the northern tip of the island in F1 (Figs. 12 and 13d), roughly from 1600 UTC 1 to 0300 UTC 2 June. Thus, the intense rainfall is more concentrated in a smaller area, and a considerably higher 24-h peak amount of 541 mm is achieved at the northern coast of Taiwan in F1. Note also that in Fig. 12, only few other rainbands exist with comparable intensity nearby than the one responsible for the coastal rainfall in northern Taiwan.

**5.4 Frontal disturbance and its relation to rainbands**

In Fig. 12 where the rainband is fixed in location for many hours, a slow-moving frontal disturbance is also visible to develop over the northern Taiwan Strait, to the northwest of Taiwan since 1400 UTC and until at least 2200 UTC 1 June (Figs. 12a-e). As the westerly flow to the south of its cyclonic center is enhanced, it appears to produce stronger near-surface convergence with the southwesterly flow off northwestern Taiwan, and subsequently with the northeasterly flow off northern

and northeastern Taiwan in F1. To further examine this linkage, the pressure, horizontal wind, and convergence fields in F1 at the height of 575 m are shown in Fig. 14, at the same times as in Fig. 12 for comparison.

In Fig. 14, the low-pressure disturbance along the Mei-yu front forms before 1400 UTC 1 June and is still identifiable at 0000 UTC 2 June (Figs. 14a-f). From 1400 to at least 1800 UTC (Figs. 14a-c), narrow but intense convergence lines (over $10^{-3}$ s$^{-1}$ in magnitude) near the surface develop not only near the northwestern coast over the northern Taiwan Strait (zone 1),

between the westerly flow in the southern quadrant of the low and the southwesterly flow deflected and enhanced by the topography of Taiwan further south, but also off the northern tip of the island, in a northwest-southeast orientation (zone 2) between the westerly flow and the post-frontal northeasterly winds. After 1800 UTC, as the low moves south together with the front in F1, the westerly and southwesterly flows merge together and form the convergence with the northeasterly flow as





a continuation of zone 2 for about 6 h until 0000 UTC 2 June (Figs. 14d-f). Note that throughout this time, especially after
1800 UTC, zone 2 is stationary (in its southern part) and extends right into the northern coast of Taiwan, near the location of
maximum rainfall in the observation (cf. Fig. 6a). Its location corresponds well with the persistent rainband (cf. Fig. 12), and
is clearly contributed to the maximum rainfall which is also aligned from northwest to southeast across the northern coast (cf.
Fig. 9b). Little rainfall exists near the low center, so that the offshore region to the northwest of northern Taiwan receives
much less rainfall in F1 compared to S1 (cf. Figs. 9 and 12). After 0000 UTC 2 June, the low disturbance in F1 weakens and
the convergence zone also slowly moves south (Figs. 14g-i), gradually away from the northern coast of Taiwan, and so is the
rainband shortly thereafter (Fig. 12). Therefore, in F1, this frontal low-pressure disturbance is identified as the key feature
that enhanced the westerly or southwesterly flow, and more importantly, produced a stationary near-surface convergence
zone with the postfrontal flow right across the northern coast of Taiwan for several hours, thus leading to the persistent
rainband also fixed in location. Both the frontal disturbance and a stationary convergence zone like zone 2 do not appear in
S1, and subsequently its rainbands are all migratory and the rainfall is not as concentrated (cf. Figs. 11 and 13a-c).

## 6 Discussion

While the near-surface convergence associated with the rainbands in S1 (figure omitted) does not appear to be weaker than
those in F1 (Fig. 14) by comparison, the frontal intensity is the only factor yet to be discussed among the five features
identified in WLC21, and thus perhaps should be addressed here. In Fig. 15, the equivalent potential temperature ($\theta_e$) fields
at 575 m at three selected times during the heavy-rainfall period in S1 and F1 are shown and compared. In S1, the
convergence near the wind-shift line is only directly between the southwesterly flow over the Taiwan Strait and the
postfrontal northeasterly flow farther to the north as discussed, but often possesses a $\theta_e$ difference of at least 10-12 K across
a distance of about 0.5° (~55 km) near northern Taiwan (Figs. 15a-c). On the other hand, with the appearance of the frontal
low in F1, the westerly flow to its south that converges and later merges with the southwesterly flow is typically colder in $\theta_e$
(Figs. 15d-f), as it is partially circulated from the colder air to the west of the low center. As the $\theta_e$ south of the wind-shift
line is lower in value (typically below 342 K), the contrast across the line is also less and only about 5-6 K. Therefore, the
low-pressure disturbance in F1 does not bring about a larger $\theta_e$ contrast across the front (and rainband), and the higher peak
rainfall amount along the northern coast of Taiwan can be confirmed to be mainly due to the persistent rainband fixed in
location in our 1-km experiment.

As the near-surface frontal disturbance over the northern Taiwan Strait in F1 is identified to be the key feature that leads to
the persistent convergence zone and rainband fixed in location across the northern coast of Taiwan, and thus the
considerably higher peak amount there (541 mm in 24 h), compared to the S1 experiment (393 mm). One may ask if there
any observational evidence to support the presence of the low? To address this point, the CWB regional weather charts every
3 h during the heavy-rainfall period are presented in Fig. 16. While the frontal position (analyzed in this study) in the
regional chart often differs from the synoptic map (cf. Fig. 2) as expected, a frontal low of about 999-1001 hPa in mean sea-



level pressure (MSLP) is seen to appear about 150 km north of Taiwan as early as 0600 UTC (Fig. 16a), and move slowly eastward along the front as the latter gradually approached Taiwan until one day later (Figs. 16b-i). Throughout this period, the MSLP at its center was consistently about 1-3 hPa lower than the surrounding, and the only time at least one enclosed isobar could not be identified is 0300 UTC 2 June (Fig. 16h). Thus, the near-surface low along the front, during the entire

time when it is captured in F1 (cf. Fig. 14), is confirmed to exist in the observation and quite persistent as well.

**7 Conclusion and summary**

In this study, the extreme rainfall event on 2 June 2017 in northern Taiwan, where the peak 24-h amount of 645 mm was observed over the coast, is studied through numerical modeling. In an earlier study, WLC21 employed ensemble sensitivity analysis to identify some factors important to differentiate more-rainy (around 150-350 mm) from less-rainy (< 150 mm)

members at grid sizes of 2.5-5 km: including the moving speed of the surface front (and 700-hPa wind-shift line), moisture amount near the front, location and timing of frontal disturbance, and frontal intensity. Following WLC21, two experiments in this study at a finer grid size of 1 km produced a peak amount of 541 (exp. F1) and 393 mm (exp. S1), respectively, and therefore are compared to isolated the reasons in F1 for its considerably higher peak amount at the northern coast of Taiwan, if and when an amount in better agreement with the observation is to be captured. The F1 run also confirms that it is possible

to reproduce the extreme heavy rainfall of ≥ 500 mm at the northern coast for this event. Besides the main objectives stated above, the topographic effects of Taiwan on rainfall in this event is also examined and tested using three 3-km experiments.

In S1 where the peak 24-h rainfall is less (nearly 400 mm) at the northern coast, the surface front has stronger contrast in $\theta_e$ and moves slightly slower, and the convergence is of similar strength compared to F1, so these factors are not crucial in raising the peak rainfall to beyond 400 mm. Its rainfall is more widespread over a larger area, produced by several slow-

moving yet migratory rainbands through the northernmost part of Taiwan. On the contrary, in F1 where the peak rainfall is higher and reaches 541 mm overland (and 618 mm nearshore), the responsible rainband remains stationary across the coastline over an extended period, in good agreement with the observation, and is caused by the convergence between the southwesterly flow and the colder northeasterly flow behind the front. A frontal low-pressure disturbance to the northwest of northern Taiwan is identified in F1 to lead to westerly flow to its south that combines with the topographically-deflected

southwesterly flow, and the subsequent convergence (at the leading edge) with the post-frontal flow for much of the heavy-rainfall period. With the rainband fixed in location, the rainfall is more concentrated and a higher peak amount is achieved. Confirmed in observation, this near-surface frontal disturbance does not exist in S1.

For the topographic effect, our 3-km experiment and sensitivity tests indicate significant differences when the entire island topography of Taiwan is removed. Without the blocking and deflecting effects on the pre-frontal flow, there is no heavy

rainfall in northern Taiwan. However, the Datun Mountain in northern Taiwan, when removed, produces only minor differences in rainfall, so its impacts are smaller here compared to those recently obtained by Tu et al. (2022), where the reduction in peak rainfall is estimated to be around 25%. This result at least indicates that enough variability and uncertainty exist among model simulations to draw a definite conclusion on the importance of Datun Mountain in this event.





Although the peak amount in S1 simulation is less, the F1 does produce 541 mm on land and only about 100 mm below the
observation, driven by M18 that was completed well before the occurrence of the actual event (roughly 60 h prior). Thus, the
1-km forecast offers some hope to successfully predict the event in advance in real time. Some related work is currently
underway and will be reported in the future.

**8 Data availability**

The      CReSS      model      and      the      user's      guide      are      available      at      http://www.rain.hyarc.nagoya-
u.ac.jp/~tsuboki/cress_html/index_cress_eng.html.      The      NCEP      GFS      analysis/forecast      data      are      from
http://rda.ucar.edu/datasets/ds335.0/#!description,      and      the      NAVGEM      data      are      from
https://www.hycom.org/dataserver/navgem. The observational data in Taiwan are from the CWB (https://cwb.gov.tw/) and
the DBAHR (https://dbar.pccu.edu.tw/).

*Acknowledgements.* The authors thank the reviewers for their constructive comments that helped improve the manuscript.
Useful discussions with Profs. Yu-Chieng Liou (National Central University) and Ben Jong-Dao Jou (National Taiwan
University) are appreciated. The various data used in this study are provided by the CWB, DBAHR, the National Science
and Technology Center for Disaster Reduction (NCDR) of Taiwan, and NCEP and the Center for Ocean-Atmospheric
Prediction Studies (COAPS) of the USA. This study is supported by the Ministry of Science and Technology of Taiwan,
under grants MOST 108-2111-M-003-005-MY2 and MOST 110-2111-M-003-004.

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



**Table 1.** The physical package used by all CReSS experiments (with references) in this study.

| | |
|---|---|
| Cloud microphysics | Double-moment bulk cold-rain (Lin et al., 1983; Cotton et al., 1986; Murakami, 1990; Ikawa and Saito, 1991; Murakami et al., 1994) |
| PBL turbulence | 1.5-order closure with prediction of turbulent kinetic energy (Deardorff, 1980; Tsuboki and Sakakibara, 2007) |
| Surface processes | Energy/momentum fluxes, shortwave and longwave radiation (Kondo, 1976; Louis et al., 1982; Segami et al., 1989) |
| Substrate model | 43 levels, every 5 cm to 2.1 m (Tsuboki and Sakakibara, 2007) |





**Table 2.** The domain configuration, IC/BCs, simulation period (UTC), and other relevant settings of three CReSS experiments (S3, S1, and F1) in this study. For grid configuration, the numbers are in $x \times y \times z$.

| Experiment name | S3 (3 km) | S1 (1 km) | F1 (1 km) |
|---|---|---|---|
| Projection | Lambert conformal (secant at 10° and 40°N, center at 120°E) | | |
| Grid size (km) | 3.0 × 3.0 × 0.2-0.624 (0.5)* | 1.0 × 1.0 × 0.1-0.681 (0.5)* | |
| Grid dimension | 1152 × 672 × 52 | 840 × 600 × 50 | |
| Domain size (km) | 3456 × 2016 × 26 | 840 × 600 × 25 | |
| IC/BCs | NCEP GFS FNL analyses (0.25° × 0.25°, 6 h) | Exp. S3 | M18 of WLC21 |
| Simulation period | 0000 UTC 30 May to 0000 UTC 4 Jun 2017 (120 h) | 2200 UTC 31 May to 0400 UTC 2 Jun 2017 (30 h) | 1300 UTC 1 Jun to 1900 UTC 2 Jun 2017 (30 h) |
| Output frequency (h) | 1 h | 1 h | |

* The vertical grid spacing of CReSS is stretched (smallest at bottom), and the parentheses give the averaged value.



**Table 3.** Design and brief description of the five CReSS experiments included in this study. Member M18 of WLC21 has the same setting as S3 and was driven by the NCEP GFS gross analyses and forecasts ($0.5° \times 0.5°$, 26 levels, every 6 h).

| | |
|---|---|
| 3-km experiments | |
| S3 | Control simulation at 3-km grid size (cf. Table 2) |
| S3-NT | Identical to S3, except that the topography of Taiwan is removed |
| S3-NNT | Identical to S3, except that the topography of northern Taiwan is removed |
| 1-km experiments (cf. Table 2) | |
| S1 | Control simulation at 1-km grid size, driven by S3 with $t_0$ at 2200 UTC 31 May 2017 (for 30 h) |
| F1 | Forecast experiment at 1-km grid size, driven by M18 of WLC21 with $t_0$ at 1300 UTC 1 Jun 2017 (for 30 h) |





**Table 4.** Comparison of areal-averaged 24-h rainfall (mm) inside the three domains, denoted as large (L) domain (24.85°-25.65°N, 120.75°-122.15°E), middle (M) domain (25.05°-25.45°N, 121.1°-121.8°E), and small (S) domain (25.1°-25.3°N, 121.35°-121.8°E), respectively, in experiments S1 and F1 during their selected 24-h period (starting from 0000 UTC 1 Jun for S1 and 1300 UTC 1 Jun for F1). The three domains are depicted in Fig. 9, and the mean rainfall values are given for the full domain and land only.

| Domain | Domain L | | Domain M | | Domain S | |
|---|---|---|---|---|---|---|
| | Full | Land only | Full | Land only | Full | Land only |
| S1 | 219.86 | 179.50 | 259.88 | 221.81 | 251.32 | 242.57 |
| F1 | 213.42 | 224.27 | 330.70 | 317.48 | 346.36 | 359.54 |

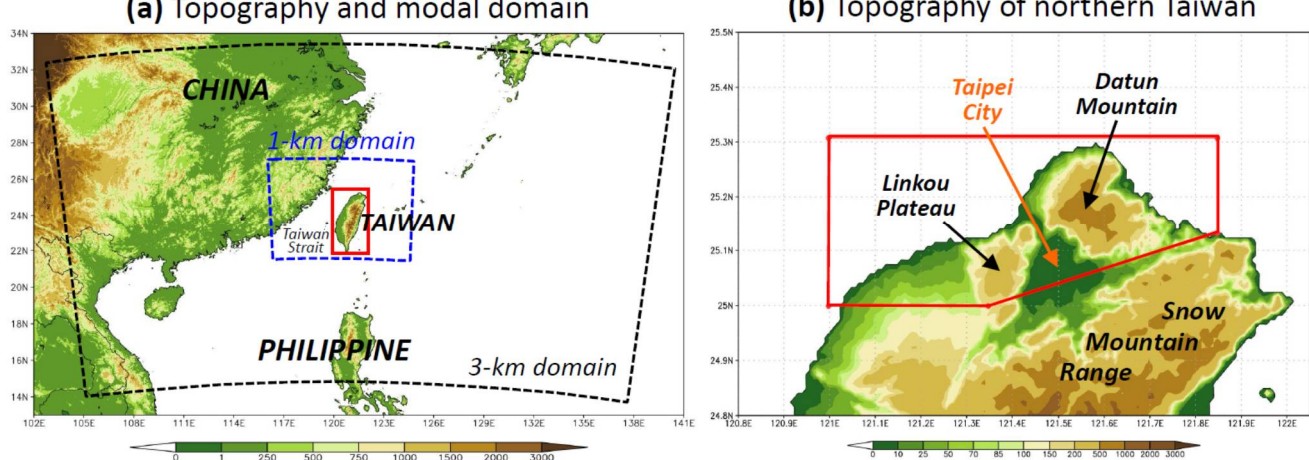

**Figure 1. (a) The topography of Taiwan and surrounding areas (m, color, scale at bottom), and the domains of 3-km and 1-km CReSS experiments. The red box depicts the region of terrain removal in the S3-NT experiment. (b) Topography (m) and the region of terrain removal in northern Taiwan used in the S3-NNT experiment (pedagon enclosed by red lines).**



**Figure 2.** The CWB surface weather charts, overlaid with the NAVGEM 925-hPa flow field, surrounding Taiwan every 6 h from (a) 1200 UTC 1 to (d) 0600 UTC 2 Jun 2017. In the panels, the mean sea-level pressure (MSLP, hPa) are analyzed with isobars every 4 hPa (thickened at 1000 hPa), and the frontal position (and type) and closed high/low centers (labelled as H/L) are marked (source: CWB and DBAHR)






**Figure 3. The CWB upper-air charts surrounding Taiwan at (a) 850, (b) 700, (c) 500, and (d) 200 hPa at 1200 UTC 1 Jun 2017. In the panels, geopotential height (gpm, solid isopleths) and temperature (°C, thin red dashed isotherms) are analyzed at intervals of 30, 30, 60, and 120 gpm, and 3°C, 3°C, 5°C, and 5°C, respectively, following the order (source: CWB). The thick red dashed lines mark troughs or wind-shift lines.**



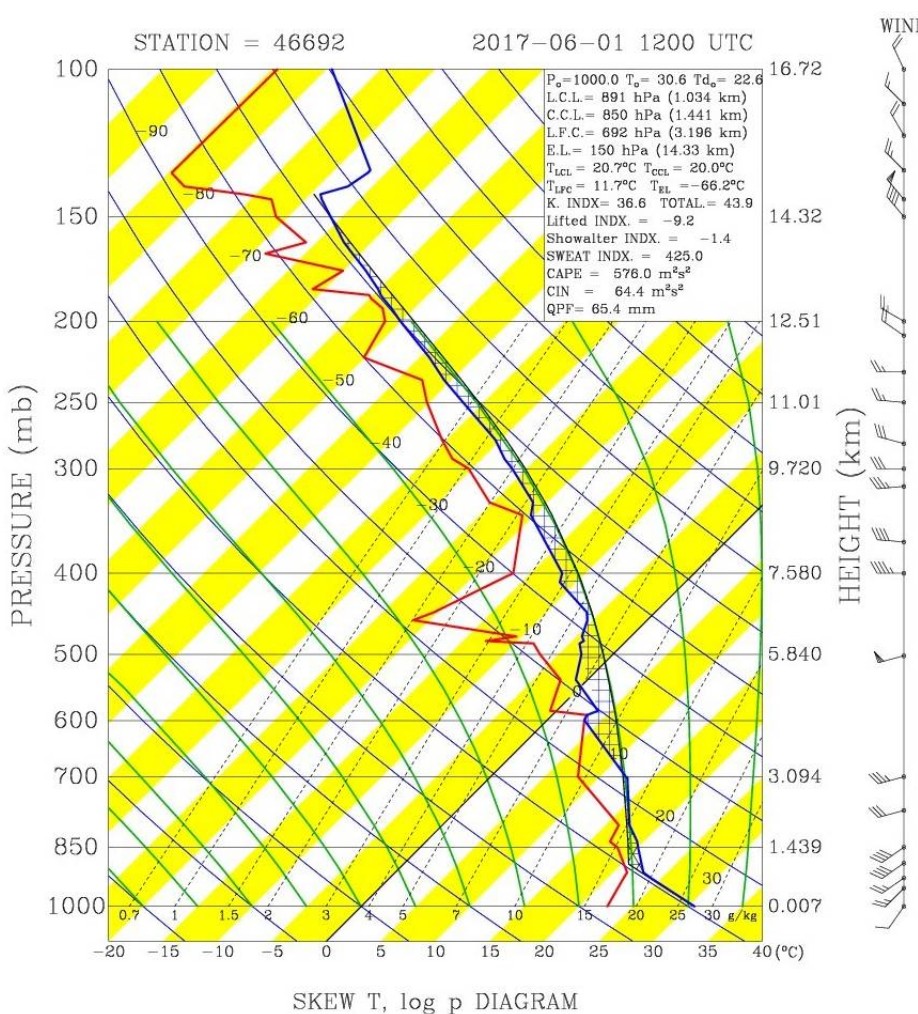

**Figure 4. The sounding and horizontal wind profiles at Panchiao, Taipei (46692) launched at 1200 UTC 1 Jun 2017. Some relevant parameters are given to the top right.**









**Figure 5. Hourly rainfall (mm) from the Quantitative Precipitation Estimation and Segregation using Multiple Sensors (QPESUMS), a radar-derived product merged with rain-gauge observations [source: CWB and the National Science and Technology Center for Disaster Reduction (NCDR) of Taiwan], overlaid with surface horizontal winds (barbs, 1 full barb = 10 m s⁻¹) in the NCEP FNL analyses, surrounding Taiwan from (a) 1800 UTC 1 to (l) 0500 UTC 2 Jun 2017. The wind fields between the 6-hourly FNL analyses are linearly interpolated in time.**


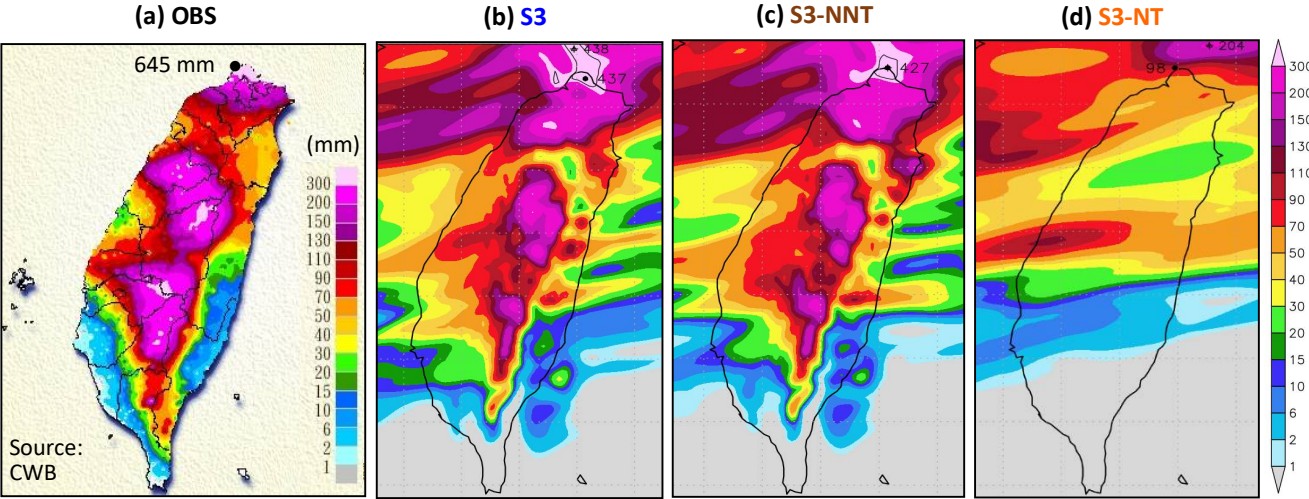

**Figure 6. Distribution of 24–h accumulated rainfall (mm, color) in Taiwan (a) from the rain-gauge observations from 1600 UTC 1 to 1600 UTC 2 Jun 2017 (source: CWB), and from three model experiments of (b) S3, (c) S3-NNT, and (d) S3-NT, respectively, all from 0700 UTC 1 to 0700 UTC 2 Jun. The maximum values overland in northern Taiwan and those offshore in model experiments are labeled (dots).**




**Figure 7.** Model surface winds at 10-m height (m s⁻¹, reference vector at bottom), frontal position (thick dashed line), and column-maximum mixing ratio of precipitating hydrometeors (rain + snow + graupel, g kg⁻¹, color) near Taiwan in the S3 experiment at (a) 0700, (b) 1300, and (c) 1900 UTC 1 Jun 2017. As in (a)-(c), except in experiment (d)-(f) S3-NNT and (g)-(i) S3-NT, respectively. The model time (h) is also labeled in each panel.



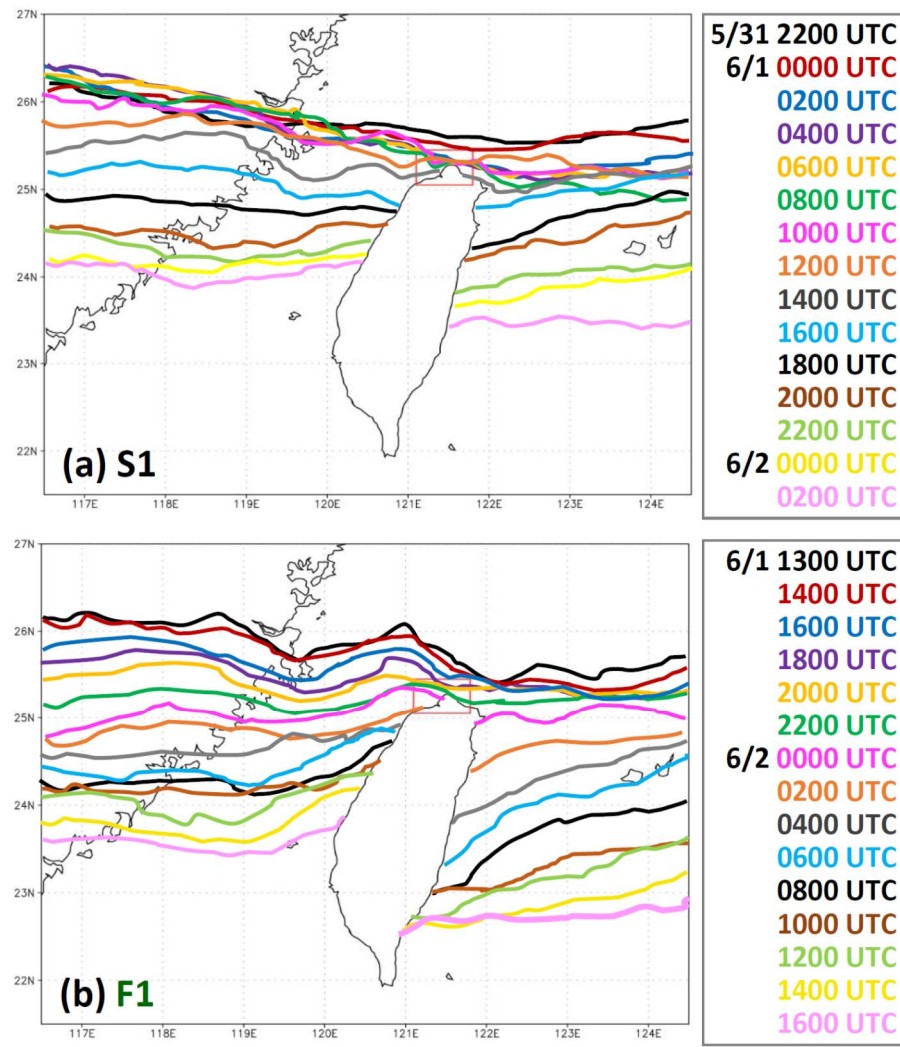

**Figure 8. Model-simulated surface frontal positions every 2 h (a) from 2200 UTC 31 May to 0200 UTC 2 Jun in S1 experiment, and (b) from 1300 UTC 1 to 1600 UTC 2 Jun 2017 in F1 experiment (additional position at 1300 UTC 1 Jun), respectively.**



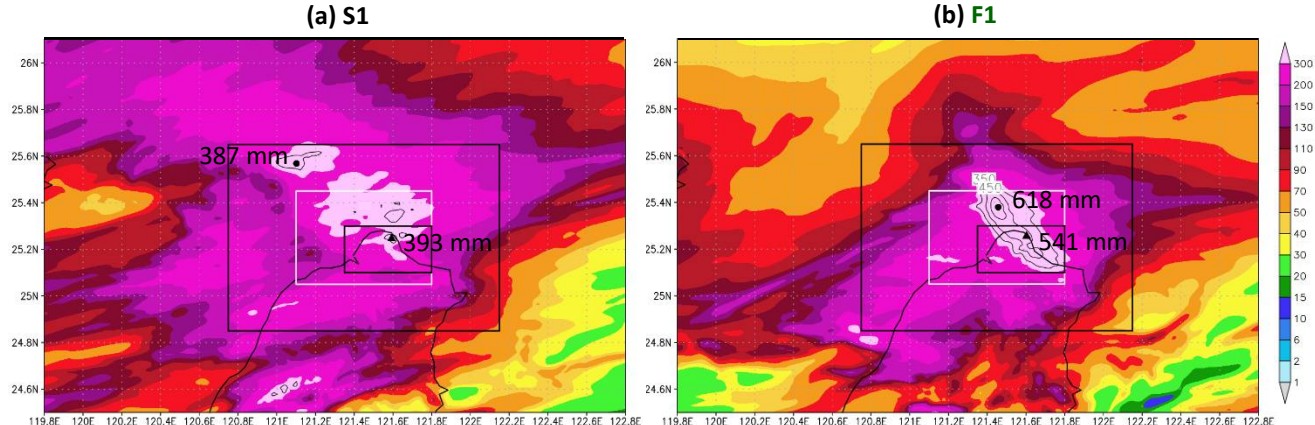

**Figure 9. Distribution of 24-h accumulated rainfall (mm, color) around northern Taiwan in experiment (a) S1 (from 0000 UTC 1 to 0000 UTC 2 Jun) and (b) F1 (from 1300 UTC 1 to 1300 UTC 2 Jun 2017), respectively. Above 300 mm, additional coutours are drawn at 350, 450, and 550 mm. The maximum values overland in northern Taiwan (triangles) and those offshore (dots) are labeled, and the three averaging domains used in Fig. 10 are also plotted.**



**Figure 10. Fractional distributions of different rainrate ranges (mm h⁻¹, see legend) inside the large (L) domain (24.85°-25.65°N, 120.75°-122.15°E) in experiment (a) S1 and (b) F1, respectively, during their selected 24-h period (starting from 0000 UTC 1 Jun for S1 and 1300 UTC 1 Jun for F1). (c),(d) and (e),(f) As in (a),(b), except for (c),(d) the middle (M) domain (25.05°-25.45°N, 121.1°-121.8°E) and (e),(f) the small (S) domain (25.1°-25.3°N, 121.35°-121.8°E), respectively. The three domains are shown in Fig. 9, and the arrival (A) and departure time (D) of the surface front in the northernmost part of Taiwan (determined using Fig. 8) are marked by vertical arrows.**




**Figure 11. Model surface winds at 10-m height (m s⁻¹, reference vector at bottom) and hourly rainfall (mm, color) near northern Taiwan every 2 h from (a) 0600 to (i) 2200 UTC 1 Jun 2017 in the S1 experiment.**





**Figure 12. Model surface winds at 10-m height (m s⁻¹, reference vector at bottom) and hourly rainfall (mm, color) near northern Taiwan every 2 h from (a) 0600 to (i) 2200 UTC 1 Jun 2017 in the F1 experiment.**



**Figure 13. Model-simulated surface rainband positions (at leading edge) around northern Taiwan at 1-h intervals during (a) 0600-1000 UTC, (b) 1200-1700 UTC, and (c) 1800-2200 UTC 1 Jun in S1 experiment, and (d) from 1400 UTC 1 to 0500 UTC 2 Jun 2017 in F1 experiment, respectively. Dashed lines repesent other rainbands nearby at the same time.**



**Figure 14. Model pressure (hPa, isobars every 2 hPa), horizontal winds (m s⁻¹, reference vector at bottom), and convergence (10⁻⁴ s⁻¹, color, divergence omitted) at the height of 575 m near northern Taiwan every 2 h from (a) 1400 UTC 1 to (i) 0600 UTC 2 Jun 2017 in the F1 experiment. The height contours at 575 m are also plotted over land (gray), and convergence zone 1 and 2 (see text for details) are also marked in (c).**



**Figure 15. Model pressure (hPa, isobars every 2 hPa), horizontal winds (m s⁻¹, reference vector at bottom), and equivalent potential**
**temperature ($\theta_e$, K, color) at the height of 575 m near northern Taiwan at (a) 0600, (b) 1000, and (c) 1600 UTC 1 Jun 2017 in the S1**
**experiment. (d)-(f) As in (a)-(c), except at (d) 1400 and (e) 1800 UTC 1, and (f) 0000 UTC 2 Jun in the F1 experiment. The height**
**contours at 575 m are plotted over land (gray), and the wind-shift lines are also depicted (thick dashed lines).**





**Figure 16. The CWB surface regional weather charts near Taiwan every 3 h from (a) 0600 UTC 1 to (i) 0600 UTC 2 Jun 2017. In the panels, the mean sea-level pressure (MSLP, hPa) are analyzed with isobars every 2 hPa with closed high/low centers labeled**
**(source: CWB), and the surface frontal position is also marked (thick dashed lines).**