# Peer review of "A modelling study of an extreme rainfall event along the northern coast of Taiwan on 2 June 2017"

_Atmospheric Chemistry and Physics, 2022_

## Referee Comment (RC1)

Wang et al. 2022, "A modelling study of an extreme rainfall event along the northern coast of Taiwan on 2 June 2017"

**Summary:**

This study examines the mechanisms responsible for producing extreme rainfall during the 2 June 2017 Mei-Yu front case in Taiwan using a series of modeling experiments. The authors study the impacts of terrain in the 3-km simulations, which were found to have negligible effects on rainfall when the northern terrain was removed, but large impacts when the whole terrain of Taiwan was removed. For this reason, they postulate that the terrain in northern Taiwan was not the main factor responsible for producing heavy rainfall. Next the authors compared two 1-km simulations–one driven by a successful ensemble member from Wang et al. (2021) and the other using the control 3-km simulation as a driver. They found that the F1 simulation produced rainfall amounts closest to observations (and closer to those previously obtained in other studies) due to a persistent rainband that was produced by a frontal disturbance that was not seen in S1. Overall, this was an interesting and well structured paper that only needs minor revisions.

**General comments:**
- How did you choose the microphysics scheme used in your study? Was it based on similar studies? And did you perform any sensitivity tests on how the microphysics scheme impacted your results?

**Specific comments**
- Line 55: Can you specify have much economic loss? It would helpful to have a number here.
- Lines 212-215: Did you specifically look at the relative height of cold hair behind the front to topography in your study? If not, please specify that this is a speculation and not actually examined.
- Lines 352–354: It is possible to compare the convergence values in observations to F1 to see if these values are similar? If so, that would be helpful to include.
- Figure 8: This is a nice figure but I suggest using a colormap that is more intuitive to distinguish early to late times. Something like using all shades of one color or cool to warms would work.
- Figure 13: This is an interesting figure but I had a hard time deciphering it. Why do you have two panels showing the same date for F1? How does the second plot below d) differ?

**Technical corrections**
- Lines 32–38: This is a long and confusing sentence. I recommend breaking this up into two separate sentences, such as "Under such conditions organized mesoscale convective (MCSs) systems such as squall lines can develop near the front and make landfall in Taiwan. The steep topography…"

- Line 53: Please remove "…the criterion to have one day off work and school" as it is not necessary information.
- Line 87: Please start a new paragraph at "A few questions remain…"
- Line 345: Please remove "As"

---

## Referee Comment (RC2)

Manuscript title:

A modelling study of an extreme rainfall event along the northern coast of Taiwan on 2 June 2017

Authors:

Chung-Chieh Wang, Ting-Yu Yeh, Ming-Siang Li, Kazuhisa Tsuboki, Ching-Hwang Liu

Summary:

Authors designed 5 cloud model runs to simulate and discuss an extreme rainfall event along the northern coast of Taiwan on 2 June 2017. The 24-h rainfall maximum along the northern coast is well simulated (541 mm) in the F1 experiment (1-km run) as compared with the rain gauge observation (645 mm). Analyses on mei-yu frontal movement and the roles of frontal disturbance are valuable. There are some major comments about the model experiments. I would suggest authors to modify the paper according to my comments to make the manuscript more complete and solid. *The paper could be publishable in Atmospheric Chemistry and Physics with major revisions.*

Major comments:

1. In the S3 experiment (3-km experiment), the surface front arrived at northern Taiwan too early, by about 9 h. Authors state that this situation is acceptable for the third day simulation. However, the arrival time error of surface front is too large (~ 9h). It means that model failed to simulate the large-scale/mesoscale weather patterns in real atmosphere (including circulations, radiation, thermodynamic processes and etc.). Thus, it is not acceptable to use the simulation of S3 to analyze the frontal characteristics. Also, one important question is that: if the authors know that there are great errors for the third-day simulation, why do author use the third-day simulation to analyze the front/frontal rainband characteristics? I strongly suggest that authors

compare model simulated 24-h rainfall accumulation and rain gauge observation from CWB by presenting the same time period during 1600 UTC 1 June to 1600 UTC 2 June (0000-2400 LST 2 June) [Figs. 6 and 9b].

2. To test and clarify the role played by the topography on the mei-yu front, authors remove topography of Taiwan (and northern Taiwan) in S3 experiment, referring to S3-NT and S3-NNT experiments. Since the frontal arrival time error is about 9 h, it is not appropriate to use S3, S3-NT, and S3-NNT experiments (3-km experiments) to discuss the interactions between mei-yu frontal system and topography over Taiwan. Instead, authors should use F1 experiment (1-km experiment; the best simulation of frontal arrival time and propagation speed in this manuscript) as the CTRL run and design the sensitivity tests of topography based on F1 experiment (e.g., F1-NT and F1-NNT).

Minor comments:

1. Line 219-221: "This is because without the terrain, the near-surface (and low-level) southwesterly winds can blow across the flattened island without the blocking effect (Figs. 7g-i),"
   should be "… (Figs. 7g-i)."
2. Line 275: delete "say,"
3. Section 7 (Line 380-383) and Figure 1b: For the "NNT" (remove northern Taiwan) run, do authors remove Datun Mountain only in the model or both Datun Mountain and Linkou Plateau? Please specify.

---

## Author Comment (AC1)

ACP-2022-377

**Reply to Reviewer 1**

**Date:** 7 Oct 2022

**Title:** A modelling study of an extreme rainfall event along the northern coast of Taiwan on 2 June 2017

**Authors:** Chung-Chieh Wang *et al.*

**Reviewer's Comments:**

**Summary:**

This study examines the mechanisms responsible for producing extreme rainfall during the 2 June 2017 Mei-Yu front case in Taiwan using a series of modeling experiments. The authors study the impacts of terrain in the 3-km simulations, which were found to have negligible effects on rainfall when the northern terrain was removed, but large impacts when the whole terrain of Taiwan was removed. For this reason, they postulate that the terrain in northern Taiwan was not the main factor responsible for producing heavy rainfall. Next the authors compared two 1-km simulations–one driven by a successful ensemble member from Wang et al. (2021) and the other using the control 3-km simulation as a driver. They found that the F1 simulation produced rainfall amounts closest to observations (and closer to those previously obtained in other studies) due to a persistent rainband that was produced by a frontal disturbance that was not seen in S1. Overall, this was an interesting and well structured paper that only needs minor revisions.

**Reply:** The positive view and constructive comments from this reviewer (Reviewer 1) are deeply appreciated. Following the instruction from the journal editor, herein we respond to all referee comments (RCs) and describe how the revision is to be performed. The actual revision will start immediately after we receive the go ahead from the editor. Below, the point-by-point responses to each of the comments from this reviewer are given, following the original comments.

**General comments:**

- How did you choose the microphysics scheme used in your study? Was it based on similar studies? And did you perform any sensitivity tests on how the microphysics scheme impacted your results?

**Reply:** The CReSS model is developed by a single research group, as described in our paper. So, for each of the physical options, it does not have many different choices, but only a few of them at different levels of complexity. This is in contrast to "community models" such as the WRF model, for which different groups can develop their own scheme and be included as an option. For cloud microphysics, the CReSS model (ver.3.4.2) has five options: no cloud microphysics, the warm-rain scheme (no ice phase), and single-moment cold-rain

scheme (both liquid and ice phases), 1.5-moment cold-rain scheme (double-moment in ice, single in liquid), and double-moment cold-rain scheme (in both liquid and ice phases). For the study, we simply choose the most complicated option, as it is the closest one to the real process in the atmosphere (no test using the less complicated schemes). This was a natural choice, and was also used in our previous studies, including Wang et al. (2021: Atmos. Res.). These above points will be clarified in the revision to an adequate level.

**Specific comments:**

- Line 55: Can you specify have much economic loss? It would helpful to have a number here.

**Reply:** Yes, the amount in economic loss will be described in the revision as suggested.

- Lines 212-215: Did you specifically look at the relative height of cold hair behind the front to topography in your study? If not, please specify that this is a speculation and not actually examined.

**Reply:** No, we did not look at the depth of the cold air because of the limited impact when the topography in northern Taiwan is removed. Thus, we will specify that this is a speculation and not actually examined in the revision, as suggested.

- Lines 352–354: It is possible to compare the convergence values in observations to F1 to see if these values are similar? If so, that would be helpful to include.

**Reply:** Thank you for this suggestion. While directly comparing the near-surface convergence values in the observations (CWB local weather maps in Fig. 16) and the F1 experiment (at 1-km grid size) are not possible, we can at least compare the scale and magnitude of the low pressure disturbance along the front (the associated convergence would be similar if their size and strength are similar). So, we will include this comparison, along the lines as suggested.

- Figure 8: This is a nice figure but I suggest using a colormap that is more intuitive to distinguish early to late times. Something like using all shades of one color or cool to warms would work.

**Reply:** Thank you for this suggestion. In the revision, a different and more intuitive color scale will be used to replot Fig. 8 to better show the frontal movement, as suggested.

- Figure 13: This is an interesting figure but I had a hard time deciphering it. Why do you have two panels showing the same date for F1? How does the second plot below d) differ?

**Reply:** In this figure, two panels are plotted for some of the time periods, such as (b) and (d). The second one is intended to show newly-developed rainbands (dashed lines) at the same time as the old rainbands (solid lines). In the revision, their meaning will be clarified as suggested.

**Technical comments:**

- Lines 32–38: This is a long and confusing sentence. I recommend breaking this up into two separate sentences, such as "Under such conditions organized mesoscale convective (MCSs) systems such as squall lines can develop near the front and make landfall in Taiwan. The steep topography…"

**Reply:** Suggestion accepted.

- Line 53: Please remove "…the criterion to have one day off work and school" as it is not necessary information.

**Reply:** Suggestion accepted.

- Line 87: Please start a new paragraph at "A few questions remain…"

**Reply:** Suggestion accepted.

- Line 345: Please remove "As"

**Reply:** Suggestion accepted.

---

## Author Comment (AC2)

ACP-2022-377

**Reply to Reviewer 2**

**Date:** 7 Oct 2022

**Title:** A modelling study of an extreme rainfall event along the northern coast of Taiwan on 2 June 2017

**Authors:** Chung-Chieh Wang *et al.*

**Reviewer's Comments:**

**Summary:**

Authors designed 5 cloud model runs to simulate and discuss an extreme rainfall event along the northern coast of Taiwan on 2 June 2017. The 24-h rainfall maximum along the northern coast is well simulated (541 mm) in the F1 experiment (1-km run) as compared with the rain gauge observation (645 mm). Analyses on mei-yu frontal movement and the roles of frontal disturbance are valuable. There are some major comments about the model experiments. I would suggest authors to modify the paper according to my comments to make the manuscript more complete and solid. The paper could be publishable in Atmospheric Chemistry and Physics with major revisions.

**Reply:** The constructive comments from this reviewer (Reviewer 2) are very much appreciated. Following the instruction from the journal editor, herein we respond to all referee comments (RCs) and describe how the revision is to be performed. The actual revision will start immediately after we receive the go ahead from the editor. Below, the point-by-point responses to each of the comments from this reviewer are given, following the original comments.

**Major comments:**

In the S3 experiment (3-km experiment), the surface front arrived at northern Taiwan too early, by about 9 h. Authors state that this situation is acceptable for the third day simulation. However, the arrival time error of surface front is too large (~ 9h). It means that model failed to simulate the large-scale/mesoscale weather patterns in real atmosphere (including circulations, radiation, thermodynamic processes and etc.). Thus, it is not acceptable to use the simulation of S3 to analyze the frontal characteristics. Also, one important question is that: if the authors know that there are great errors for the third-day simulation, why do author use the third-day simulation to analyze the front/frontal rainband characteristics? I strongly suggest that authors compare model simulated 24-h rainfall accumulation and rain gauge observation from CWB by presenting the same time period during 1600 UTC 1 June to 1600 UTC 2 June (0000-2400 LST 2 June) [Figs. 6 and 9b].

**Reply:** Thank you for your comment. We agree with this reviewer that the sensitivity tests on the topography should be based on an experiment that has a better overall agreement with the

observation, if possible. Therefore, in the revision, we use the M18 experiment (and rename it to F3, meaning 3-km forecast) as the 3-km control run and analyze the characteristics of the front and frontal rainbands in this run as suggested, rather than the S3 experiment. As for the 24-h accumulation, the reason for the CWB to produce its routine rainfall maps from 1600 UTC is because it is 0000 LST in Taiwan, so Fig. 6a shows the daily rainfall. In other words, it was not based on any reason linked to this case per se. In this case, it just so happens that 1600 UTC was about one hour before the rain started in the northern coast, where the episode lasted for about 10 h as shown in Fig. 5. So we simply took the CWB plot and used it directly, without making another plot using a selected 24-h window. In Figs. 9 and 10, the rainfall in two 1-km experiments (F1 and S1) are compared. In F1, the rainfall starts right after 1600 UTC so it is spot on in the timing of rainband arrival (see Fig. 12), but we chose 1300-1300 UTC in order to show the whole event (lasting for about 12 h) to the readers clearly, with a few hours of lead time. A similar 24-h time frame was selected for S1 based on the same reason. The comparison between the two runs and with the observation is fair as long as the same length (24 h) is used and the whole rainfall episode in northern Taiwan is covered in a similar way, as in Figs. 9 and 10. In addition, the S1 experiment is used to illustrate why it had less total rainfall (or, equivalently, why F1 had more rainfall) in the model, so it is OK to exhibit larger errors including timing errors, as long as the evolution is a realistic one. In the revision, we also state that the larger timing error in S1 could be a factor of its lower rainfall accumulation along the northern coast in Taiwan, and the above points are also better clarified, along the lines as suggested.

To test and clarify the role played by the topography on the mei-yu front, authors remove topography of Taiwan (and northern Taiwan) in S3 experiment, referring to S3-NT and S3-NNT experiments. Since the frontal arrival time error is about 9 h, it is not appropriate to use S3, S3-NT, and S3-NNT experiments (3-km experiments) to discuss the interactions between mei-yu frontal system and topography over Taiwan. Instead, authors should use F1 experiment (1-km experiment; the best simulation of frontal arrival time and propagation speed in this manuscript) as the CTRL run and design the sensitivity tests of topography based on F1 experiment (e.g., F1-NT and F1-NNT).

**Reply:** Thank you for this comment. As stated above, we agree with this reviewer that the sensitivity tests on the topography should be based on the M18 experiment (which is called F3 now), and two new terrain-removal experiments (F3-NT and F3-NNT, respectively) had been performed for sensitivity tests. The differences are similar to what we had in our previous draft (among S3, S3-NT, and S3-NNT), as shown below in Fig. B1. In F3-NNT, the northern coast does not receive less total rainfall (compared to F3) when the northern terrain is removed.

[Figure]

**Fig. B1.** Total 24-h accumulated rainfall (mm) surrounding Taiwan in (a) F3, (b) F3-NNT, and (c) F3-NT experiments, respectively.

Right now, we are also running F1-NT and F1-NNT experiments driven by F3-NT and F3-NNT, respectively to confirm our results on topography, as suggested by this reviewer. We expect them to give similar results as F3-NT and F3-NNT. Once these experiments are finished, we will use them in the revision, as suggested. We may still show the 3-km tests in our revision for comparison, as this is the model resolution comparable to Tu et al. (2022) in their tests (their Fig. 18).

**Minor comments:**

1.  Line 219-221: "This is because without the terrain, the near-surface (and low-level) southwesterly winds can blow across the flattened island without the blocking effect (Figs. 7g-i)," should be "… (Figs. 7g-i)."

**Reply:** Suggestion accepted.

2.  Line 275: delete "say,"

**Reply:** Suggestion accepted.

3.  Section 7 (Line 380-383) and Figure 1b: For the "NNT" (remove northern Taiwan) run, do authors remove Datun Mountain only in the model or both Datun Mountain and Linkou Plateau? Please specify.

**Reply:** Both of them are removed. In the revision, this will be better clarified as suggested.

---

## Author Response (AR1)

ACP-2022-377

**Reply to Reviewer 1**

**Date:** 25 Nov 2022

**Title:** A modelling study of an extreme rainfall event along the northern coast of Taiwan on 2 June 2017

**Authors:** Chung-Chieh Wang *et al.*

**Reviewer's Comments:**

**Summary:**

This study examines the mechanisms responsible for producing extreme rainfall during the 2 June 2017 Mei-Yu front case in Taiwan using a series of modeling experiments. The authors study the impacts of terrain in the 3-km simulations, which were found to have negligible effects on rainfall when the northern terrain was removed, but large impacts when the whole terrain of Taiwan was removed. For this reason, they postulate that the terrain in northern Taiwan was not the main factor responsible for producing heavy rainfall. Next the authors compared two 1-km simulations–one driven by a successful ensemble member from Wang et al. (2021) and the other using the control 3-km simulation as a driver. They found that the F1 simulation produced rainfall amounts closest to observations (and closer to those previously obtained in other studies) due to a persistent rainband that was produced by a frontal disturbance that was not seen in S1. Overall, this was an interesting and well structured paper that only needs minor revisions.

**Reply:** The positive view and constructive comments from this reviewer (Reviewer 1) are deeply appreciated. Based on the instruction and recommendation of the journal editor, we have revised our manuscript following all referee comments (RCs) closely and prepared the authors' reply to these comments (RCs). In the revised and color-coded manuscript, modifications made in response to comments from Reviewer 1, Reviewer 2, and by ourselves (mostly minor changes and corrections) are marked in red, blue, and orange, respectively. Below, the point-by-point responses to each of the comments from this reviewer are given, following the original comments.

**General comments:**

- How did you choose the microphysics scheme used in your study? Was it based on similar studies? And did you perform any sensitivity tests on how the microphysics scheme impacted your results?

**Reply:** The CReSS model is developed by a single research group, as described in our paper. So, for each of the physical processes, it does not have many different choices, but only a few of them at different levels of complexity. This is in contrast to "community models" such as

the NCAR-supported WRF model, for which different groups can develop their own scheme and be included as an option. For cloud microphysics, the CReSS model (ver.3.4.2) has five options: no cloud microphysics, the warm-rain scheme (no ice phase), and single-moment cold-rain scheme (both liquid and ice phases), 1.5-moment cold-rain scheme (double-moment in ice, single in liquid), and double-moment cold-rain scheme (in both liquid and ice phases). For the study, we simply choose the most complicated option, as it is the closest one to the real process in the atmosphere (no test using the less complicated schemes). This was a natural choice, and was also used in our previous studies, including Wang et al. (2021: Atmos. Res.). In the revision, the above point is better clarified (L130-134), as suggested.

**Specific comments:**

- Line 55: Can you specify have much economic loss? It would helpful to have a number here.

**Reply:** The amount in economic loss has been added in the revision with a reference (L69-70, L464-465), as suggested.

- Lines 212-215: Did you specifically look at the relative height of cold hair behind the front to topography in your study? If not, please specify that this is a speculation and not actually examined.

**Reply:** No, we did not look at the depth of the cold air because of the limited impact when the topography in northern Taiwan is removed. However, because this part of the text has been revised based on new 1-km sensitivity tests on the topography following the recommendation from Reviewer 2, the depth of the cold air behind the front is no longer mentioned and the suggested change becomes not applicable.

- Lines 352–354: It is possible to compare the convergence values in observations to F1 to see if these values are similar? If so, that would be helpful to include.

**Reply:** Thank you for this suggestion. While directly comparing the near-surface convergence values in the observations (CWB local weather maps in Fig. 16) and the F1 experiment (at 1-km grid size) are not possible, we can at least compare the scale and magnitude of the low pressure disturbance along the front (the associated convergence would be similar if their size and strength are similar). So, this comparison has been included in the revision (L363-364), along the lines as suggested.

- Figure 8: This is a nice figure but I suggest using a colormap that is more intuitive to distinguish early to late times. Something like using all shades of one color or cool to warms would work.

**Reply:** Thank you for this suggestion. In the revision, a different and more intuitive color scale will be used to replot Fig. 8 to better show the frontal movement (Fig. 8 in p.31), as suggested.

- Figure 13: This is an interesting figure but I had a hard time deciphering it. Why do you have two panels showing the same date for F1? How does the second plot below d) differ?

**Reply:** In this figure, two panels are plotted for some of the time periods, such as (b) and (d). The second one is intended to show newly-developed rainbands (dashed lines) at the same time as the old rainbands (solid lines). In the revision, their meanings are clarified as suggested (L656-657 in p.36).

**Technical comments:**

- Lines 32–38: This is a long and confusing sentence. I recommend breaking this up into two separate sentences, such as "Under such conditions organized mesoscale convective (MCSs) systems such as squall lines can develop near the front and make landfall in Taiwan. The steep topography…"

**Reply:** Suggestion accepted (L34-39).

- Line 53: Please remove "…the criterion to have one day off work and school" as it is not necessary information.

**Reply:** Suggestion accepted (L53-54).

- Line 87: Please start a new paragraph at "A few questions remain…"

**Reply:** Suggestion accepted (it was already a new paragraph here in the previous draft, L88).

- Line 345: Please remove "As"

**Reply:** Suggestion accepted (L353).

**Reply to Reviewer 2**

**Date:** 25 Nov 2022

**Title:** A modelling study of an extreme rainfall event along the northern coast of Taiwan on 2 June 2017

**Authors:** Chung-Chieh Wang *et al.*

**Reviewer's Comments:**

**Summary:**

Authors designed 5 cloud model runs to simulate and discuss an extreme rainfall event along the northern coast of Taiwan on 2 June 2017. The 24-h rainfall maximum along the northern coast is well simulated (541 mm) in the F1 experiment (1-km run) as compared with the rain gauge observation (645 mm). Analyses on mei-yu frontal movement and the roles of frontal disturbance are valuable. There are some major comments about the model experiments. I would suggest authors to modify the paper according to my comments to make the manuscript more complete and solid. The paper could be publishable in Atmospheric Chemistry and Physics with major revisions.

**Reply:** The constructive comments from this reviewer (Reviewer 2) are very much appreciated. Based on the instruction and recommendation of the journal editor, we have revised our manuscript following all referee comments (RCs) closely and prepared the authors' reply to these comments (RCs). In the revised and color-coded manuscript, modifications made in response to comments from Reviewer 1, Reviewer 2, and by ourselves (mostly minor changes and corrections) are marked in red, blue, and orange, respectively. Below, the point-by-point responses to each of the comments from this reviewer are given, following the original comments.

**Major comments:**

In the S3 experiment (3-km experiment), the surface front arrived at northern Taiwan too early, by about 9 h. Authors state that this situation is acceptable for the third day simulation. However, the arrival time error of surface front is too large (~ 9h). It means that model failed to simulate the large-scale/mesoscale weather patterns in real atmosphere (including circulations, radiation, thermodynamic processes and etc.). Thus, it is not acceptable to use the simulation of S3 to analyze the frontal characteristics. Also, one important question is that: if the authors know that there are great errors for the third-day simulation, why do author use the third-day simulation to analyze the front/frontal rainband characteristics? I strongly suggest that authors compare model simulated 24-h rainfall accumulation and rain gauge observation from CWB by

presenting the same time period during 1600 UTC 1 June to 1600 UTC 2 June (0000-2400 LST 2 June) [Figs. 6 and 9b].

**Reply:** Thank you for your comment. We agree with this reviewer that the sensitivity tests on the topography should be based on an experiment that has a better overall agreement with the observation, if possible. Therefore, in the revision, we define the F3 and F1 experiments (where F3 is the same as M18, which drove the F1 experiment) as the control run to analyze the characteristics of the front and frontal rainbands instead of using F3, as suggested (L137-149, L201-209, L232-246, L248-251, L393-395, L569-571, Table 2 in p.20, Table 4 in p.22, L623-625, Fig. 6 in p.29, L628-630, Fig. 7 in p.30). For the 24-h accumulation, during the revision we found that a mistake was made in our previous draft: the 24-h accumulation was actually from 1600 UTC and same as in the CWB plot (since 1600 UTC is 0000 LST in Taiwan), but we mistook it as from 1300 UTC. Please accept our apology for this mistake. Subsequently, we have corrected this mistake in the revision (L259, L591, L635-636), and also used the same 24-h period for rainfall comparison in both newly-produced 1-km tests on the role of topography (see our response further below, after the next paragraph), both as suggested.

Only in Fig. 10 (in p.33), where the evolution of hourly rainfall in two 1-km experiments (F1 and S1) are compared, the starting time is set to 1300 UTC for F1 in order to show the whole event (lasting for about 12 h) clearly to the readers, with 2-3 hours of lead time. A similar 24-h time frame was selected for S1 based on the same reason. The comparison between the two runs is fair as long as the time period covers the whole rainfall episode in northern Taiwan in a similar way (as is the case in Fig. 10). In addition, the S1 experiment is used to illustrate why it had less total rainfall (or, equivalently, why F1 had more rainfall) in the model, so it is OK to exhibit larger errors including the timing error, as long as the evolution is a realistic one. In the revision, the above points are also better clarified (L141-143, L232-243, L249-251), along the lines as suggested.

To test and clarify the role played by the topography on the mei-yu front, authors remove topography of Taiwan (and northern Taiwan) in S3 experiment, referring to S3-NT and S3-NNT experiments. Since the frontal arrival time error is about 9 h, it is not appropriate to use S3, S3-NT, and S3-NNT experiments (3-km experiments) to discuss the interactions between mei-yu frontal system and topography over Taiwan. Instead, authors should use F1 experiment (1-km experiment; the best simulation of frontal arrival time and propagation speed in this manuscript) as the CTRL run and design the sensitivity tests of topography based on F1 experiment (e.g., F1-NT and F1-NNT).

**Reply:** Thank you for this comment. As stated above, we agree with this reviewer that the sensitivity tests on the topography should be based on the F1 experiment (driven by F3). Two new sets of 3-km and 1-km terrain-removal experiments (the 1-km ones are named F1-NT and F1-NNT, respectively) had been performed for sensitivity tests, as suggested by this reviewer (L100-102, L104-105, L133-134, L150-155, L210-231, L375, L582-583, Table 3 in p.21, L599-601, L623-625, Fig. 6 in p.29, L628-630, Fig. 7 in p.30). Based on these tests, the differences are similar to the response in Tu et al. (2022), so we have modified our description in the revision and stated that our test had a response in general agreement with Tu et al. (2022), along the lines as suggested (L18-21, L210-219, L264, L327, L337, L388-392). Related to the additional work, a

new co-author has also been added (L3), but nevertheless such tests on the role of topography is a minor goal of the present study.

**Minor comments:**

1.  Line 219-221: "This is because without the terrain, the near-surface (and low-level) southwesterly winds can blow across the flattened island without the blocking effect (Figs. 7g-i)," should be "… (Figs. 7g-i)."

**Reply:** Corrected as suggested (L224).

2.  Line 275: delete "say,"

**Reply:** Suggestion accepted (L282).

3.  Section 7 (Line 380-383) and Figure 1b: For the "NNT" (remove northern Taiwan) run, do authors remove Datun Mountain only in the model or both Datun Mountain and Linkou Plateau? Please specify.

**Reply:** Both of them are removed. In the revision, this has been better clarified as suggested (L155, L217, L390).